# Entropy of Badminton Strike Positions

**DOI:** 10.3390/e23070799

**Published:** 2021-06-23

**Authors:** Javier Galeano, Miguel-Ángel Gomez, Fernando Rivas, Javier M. Buldú

**Affiliations:** 1Complex System Group, Universidad Politécnica de Madrid, 28040 Madrid, Spain; 2Department of Social Sciences, Physical Activity, Sport and Leisure, Universidad Politécnica de Madrid, 28031 Madrid, Spain; miguelangel.gomez.ruano@upm.es; 3Spanish Badminton Federation, 28040 Madrid, Spain; fernandorivas77@gmail.com; 4Complex System Group & GISC, Universidad Rey Juan Carlos, 28933 Móstoles, Spain; javier.buldu@urjc.es; 5Laboratory of Biological Networks, Center for Biomedical Technology, Universidad Politécnica de Madrid, 28223 Pozuelo de Alarcón, Spain

**Keywords:** shannon entropy, spatial entropy, racket sports, match analysis, performance

## Abstract

The aim of the current study was twofold: (i) to investigate the distribution of the strike positions of badminton players while quantifying the corresponding standard entropy and using an alternative metric (spatial entropy) related to winning and losing points and random positions; and (ii) to evaluate the standard entropy of the receiving positions. With the datasets of 259 badminton matches, we focused on the positions of players’ strokes and the outcome of each point. First, we identified those regions of the court from which hits were most likely to be struck. Second, we computed the standard entropy of stroke positions, and then the spatial entropy, which also considers the order and clustering of the hitting locations in a two-dimensional Euclidean space. Both entropy quantifiers revealed high uncertainty in the striking position; however, specific court locations (i.e., the four corners) are preferred over the rest. When the outcome of each point was taken into account, we observed that the hitting patterns with lower entropy were associated with higher probabilities of winning points. On the contrary, players striking from more random positions were more prone to losing the points.

## 1. Introduction

Badminton is one of the most popular sports in the world, with more than 200 million players [1]. The factors for winning a point in badminton are multiple, revealing the complexity of the sport [2]. In particular, badminton is a high-speed sport characterized by high-intensity actions interspersed by periods of effort and pause.

Research has widely described the statistical aspects of the temporal structure of badminton matches [1,3,4]: (i) the match duration is about 45–65 min; (ii) the average duration of a point is approximately nine seconds; (iii) the number of strokes per point is about 8–10. Additionally, other studies have focused their attention on understanding the differences between these parameters when considering the context, such as gender, modalities, or situational variables (number of sets, intervals of play, quality of opponent, etc.) [5,6,7,8].

On the other hand, most of the badminton literature has focused on predicting winning and losing players using key performance indicators, such as the type of service, the type of stroke, or the temporal parameters [1,3,5,6]. However, the spatial-related variables were not addressed to identify how the space of play (where each player hits the shuttlecock) was used during badminton matches, and then how these complex behaviors may identify badminton performance profiles.

Due to the different scales of the aforementioned parameters and the complexity of the game, the need to study different time scales using methodological tools of complex systems is present [9]. In particular, entropy has been used as an indicator related to the complexity of different sports [10,11]. For example, Couceiro et al. [12] have used entropy-based techniques to assess human performance variability in three different case studies: golf, tennis, and soccer. In team sports, Silva et al. [13] detailed how different measures of entropy have been applied to the study of performance variability to uncover the interactions underlying players’ and teams’ performances. In the particular case of soccer, Martinez et al. [14] have used spatial and temporal entropy to analyze the level of randomness of football passing networks, showing different levels of entropy in different teams.

Regarding badminton, however, there is still a diversity of methodologies, including assessing the level of entropy, that has yet to be utilized. In this article, the aims are twofold: (i) to investigate the distribution of the strike positions of badminton players by quantifying the corresponding standard entropy and an alternative metric (spatial entropy) related to winning and losing points and random positions; and (ii) to evaluate the standard entropy of the receiving positions. We investigate the amount of randomness associated with the position from which a player strikes the shuttlecock. In Figure 1, we plot the heat map of the striking positions of 259 badminton matches. As we can observe, the distribution of strokes seems to have some preferred regions; however, it is not homogeneous. Therefore, with the aim of understanding how random these patterns are, we first quantify the standard entropy of the striking patterns and relate it to the probability of winning a point. Next, we use an alternative metric to evaluate the entropy, which, in this case, is more related to the spatial positions of the strokes: the spatial entropy [15]. Both entropy indicators point in the same direction: there is no difference in the entropy distributions for the winning and losing points. Finally, we present a theoretical model to evaluate the level of randomness of badminton striking positions, showing the coexistence of random forces and the preference for particular locations on the court.

## 2. Materials and Methods

### 2.1. Data

The dataset examined includes 93 women’s tournaments from 2015 to 2020 with 218,081 strokes in 259 matches. Sixty-three different players are included in this analysis. The strokes made during a match were ordered sequentially with the following information obtained per stroke:

Set;Point;Stroke number in the rally;Player who made the stroke;Zone from which the stroke was made (12 zones on each player’s side);X-axis coordinate of the stroke;Y-axis coordinate of the stroke;Score player 1;Score player 2.

The sample was composed of 259 women’s matches (video recorded by the World Badminton Federation) played during the 2015–2019 World Badminton Super Series; 2016 and 2018 European Championships; the 2016 Olympic Games; and 2015, 2017, 2018, and 2019 World Championships. Data were gathered using the Dartfish video analysis software (Dartfish, Friburgo, Switzerland) by two performance analysts (with five years of experience as elite badminton coaches) trained for this observational task. In addition, data reliability was tested for inter- and intra-observer match analyses with excellent values (Kappa, ICC, Pearson’s correlation, and TEM) using two random matches.

### 2.2. Zones

A traditional way of teaching badminton is to divide the badminton court into different hitting zones. The most extensive partitioning involves dividing the court from net to end into three zones and from right to left into four zones, as we can observe in Figure 2. As we will see, we used these zones to study their distributions and calculate the standard entropy. However, the computation of the spatial entropy disregards the zones and uses the specific x,y coordinates of each stroke.

### 2.3. Standard Entropy

A common way to measure entropy in sports is to use the standard entropy [10]. In cases in which we have to measure the variability in qualitative variables, such as the different areas from which players hit in badminton, the usual method is to use the standard entropy. This entropy can be calculated by dividing the Shannon entropy by the maximum value. In our case with 12 different striking zones, the standard entropy, *H*, is calculated as follows:(1)$$H=-\frac{\sum_{i=1}^{12}p_{i}\ln\left(p_{i}\right)}{\ln(12)},$$
where $p_{i}$ is the probability of hitting from zone *i*. Consequently, the standard entropy lies in the interval from 0 to 1. A value close to zero would indicate that all the strokes have been made from a particular position, while a value close to unity reveals no preference for the striking position.

### 2.4. Spatial Entropy

Although standard entropy is probably the most common way to quantify the entropy of a system, we used another indicator that, importantly, takes into account the fact that strokes are distributed in two-dimensional Euclidean space. From the diversity of metrics quantifying the spatial randomness of the distribution of discrete objects, we chose the methodology proposed by Clark and Evans [15], since it combines the concepts of clustering, randomness, and regularity, all of them with a straightforward interpretation within the framework of badminton. In this way, for a set of *N* points (strokes) spatially distributed over a two-dimensional space, we computed the distance $r_{i,j}$ between a point *i* and its nearest neighbor *j*. Next, we obtained the mean nearest neighbor distance $\langle r\rangle =\frac{1}{N}\sum_{i\neq j}^{N}r_{i,j}$. Note that complete spatial randomness of *N* points distributed over a surface *S* is described by a Poisson process, in which the probability density function for the nearest neighbor distance, *r*, is $p\left(r\right)=2\pi \delta re^{\left(-\pi \delta r^{2}\right)}$, with δ=N/S being the point density. If we assume that each point has the same probability of appearing at any position on the surface *S*, the expected average distance between nearest neighbors is given by $r_{ran}=\frac{1}{2}\sqrt{S/N}$ [15].

By normalizing the mean nearest neighbor distance reported by the one expected in a random distribution, we obtain the spatial entropy $H_{s}=\frac{\langle r\rangle }{r_{ran}}$, which measures how far the real distribution of strokes is from a completely random one. Interestingly, values of $H_{s}$ far from one give additional information. For example, $H_{s}$ approaching zero reveals that the mean distance to the nearest neighbors is very low, which is a consequence of the existence of clusters, i.e., small regions of the court that contain high numbers of strokes. On the contrary, values of $H_{s}$ higher than unity are related to situations in which the positions of the strokes are more separated than in a random distribution. The largest value of $H_{s}$ in a two-dimensional space occurs when points are placed in a triangular lattice arrangement leading to $H_{s}^{max}=2.149$. In this way, the closer the value of $H_{s}$ is to $H_{s}^{max}$, the more regular the distribution of strokes will be. Figure 3 shows an example of how the spatial entropy captures the spatial distribution of strokes. The first three plots (a, b, and c) have been generated numerically, promoting the striking from: (a) the four corners of the court, (b) random positions, and (c) positions maximizing the distance between them. Finally, plot (d) shows a real example. When the strokes were grouped into clusters (a) the spatial entropy was 0.462. It increased to 0.997 in the purely random case (b), and reached 1.985 when the strokes were performed over a regular lattice. At the same time, for the real case $H_{s}=0.599$, revealing a situation that lies between the clustered and the random examples.

## 3. Results

### 3.1. Distributions by Zones

To understand the importance of the hitting zones, we first studied the distribution of the number of times that players hit the badminton shuttlecock from each of the 12 zones indicated in Figure 2. In Figure 2 we present a bar graph containing the probability distribution function (PDF) of all the zones. We can see that the four corners of the court (zones 1, 9, 4, and 12) were the most visited places by the players (17%, 16 %, 13.9%, and 13% of all strokes, respectively). These results reflect the pattern structure of a badminton match. In particular, back corner zones 9 and 12 are the two zones where players hit the most. Corners forward 1 and 4 positions follow. On the other hand, region 10 has the lowest percentage of strokes (only 1.1 %).

### 3.2. Standard Entropy

Next, we calculated the standard entropy (*H*) of our datasets using the information from the zone associated with each stroke calculated for each match. To understand if the behavior of the players was different when they were winning or losing, we divided the rallies into wins and losses to compute the entropy. For each match, we allocated hitting zones based on the player who won the point and the one who lost it. Based on those data, we calculated the standard entropy and repeated it for all matches. In Figure 4, we show the probability distribution of the standard entropies for points that were won and lost, which have been calculated using Equation (Equation 1) while separating winning strokes from losing strokes. In green, we can see the entropy distributions of the positions of the winning strokes and in red the positions of losing strokes of each match.

We can observe that there is no difference between the two entropy distributions. We tested whether the mean values of the distributions are equal using a paired *t*-test with a *p*-value greater than 0.5.

On the other hand, we studied the entropy of the receiving areas. To calculate this entropy, we first obtained the two-time sequences. We calculated the numbers of times the sequences appeared in the match; for example, if we wanted to study zone 1, we calculated the sequences 1–1—a hit from zone 1 received in zone 1—and 1–2, 1–3, …, 1–12. Then, we obtained the probabilities of the sequences, and finally, we obtained, using the standard entropy equation, the entropy of the position 1, and the same for the other zones. These values gave us an idea of the variability of the zones where the badminton shuttlecock can be sent from each zone.

In Figure 5a we show the average receiving entropy of all matches. Figure 5b shows the PDF of the entropy per match of the most different zones. The zones 10 and 11 had the lowest values; the zones 9 and 12 had the highest values.

### 3.3. Modeling Badminton Entropy

To understand the meaning of the standard entropy values, we carried out a series of numerical simulations implementing two types of striking patterns: (a) hitting the shuttlecock from random positions and (b) hitting the shuttlecock only from the four corners. In this way, a purely random player hit from any zone with the same probability and a purely “4-corner” player hit from one the four corner zones (1, 4, 9, and 12). Note that, for a random pattern, the probability of hitting from any zone is $p_{ran}=1/12$, since the court is divided into L=12 zones. In the case of a “4-corner” player, the hitting probability is $p_{4c}=1/4$ in the four corner zones and $p_{4nc}=0$ in the rest of the zones of the court. Next, we investigated how strokes are distributed in a combination of these two extreme situations. We used a control parameter γ∈[0,1] to assign a probability of choosing either of the two strategies. Specifically, the probability of adopting a 4-corner pattern when a new stroke was made was γ; the probability of choosing the random behavior was 1-γ. In this way, for γ=0, all strokes were made following a random pattern. For γ=1 the shuttlecock was hit from any of the four corners (each one with equal probability $p_{4c}=0.25$). Next we carried out a Monte Carlo simulation for every value of γ in the interval [0,1], with an increment of Δγ=0.01. In this way, for each value of γ we simulated the striking patterns adopted for each of the strokes of the N=259 matches, and then we computed the corresponding standard entropy *H*. Each simulation was repeated $n_{rep}=100$ times for each value of γ, leading to an average value ${\bar{H}}_{sim}\left(\gamma \right)$ with its corresponding error. Specifically, the Monte Carlo simulation consisted of the following steps:We selected the total number of strokes $L_{i}$ of a match *i*, which was taken from the real value of the N=259 matches.We set the value of gamma, ranging from zero to one.For each stroke *i*, we obtained a random number $ran_{1}\left(i\right)$ between zero and one and we compared it with the value of gamma; if $ran_{1}\left(i\right)<=\gamma$, the stroke was made according to a four-corner pattern; if $ran_{1}\left(i\right)>\gamma$, the striking zone was selected randomly.If $ran_{1}\left(i\right)<=\gamma$ (i.e., four-corner pattern), we obtained a second random number $ran_{2}\left(i\right)$ (also between 0 and 1), which was used to decide which of the four corners was selected. Specifically: zone 1 when $0<=ran_{2}\left(i\right)<0.25$, zone 4 when $0.25<ran_{2}\left(i\right)<=5$, zone 9 when $0.5<ran_{2}\left(i\right)<=0.75$, and zone 12 when $0.75<ran_{2}\left(i\right)<=1$ (see Figure 2 for the precise locations of the zones).If $ran_{1}\left(i\right)>\gamma$ (i.e., random pattern), we obtained a second random number $ran_{2}\left(i\right)$, which was used to decide which of the twelve zones was selected. The stroke was assigned to zone *i*, with i=1,2,3,…,12 if $1/12\times \left(i-1\right)<ran_{2}\left(i\right)<=1/12\times \left(i\right)$.We repeated steps 3–5 until $L_{i}$ strokes were generated and computed the Standard Entropy *H* of the resulting zone distribution.For each match, we repeated steps 1–6 up to 100 times and obtained the average of the standard entropy and its corresponding standard deviation.We repeated steps 2–7 for all values of γ in the range [0,1], with an increment of $\Delta_{\gamma }=0.01$.

In Figure 6 we show a boxplot corresponding to the standard entropies of the simulated matches $H_{sim}\left(\gamma \right)$, which have been obtained for different values of γ between the full random and the 4-corner strategies. In each box, the central mark indicates the median, and the bottom and top edges of the box indicate the 25th and 75th percentiles, respectively. We can observe a continuous evolution in the standard entropy from a "random hitting player" to a player who only hits from the four corners. The inset contains the boxplot of the standard entropy $H_{real}$ obtained from the 259 matches. Interestingly, we can observe how the situation closest to the real case corresponds to γ=0.55 with $H_{sim}=0.89$. The average of the real matches was ${\bar{H}}_{real}=0.894$.

One of the advantages of using our toy model is that we cam easily extract the analytical expression describing the value of the standard entropy H(γ) as a function of γ. The probability of striking from any of the four corners is:(2)$$p_{4c}=\gamma \frac{1}{4}+\left(1-\gamma \right)\frac{1}{12}=\frac{1+2\gamma }{12}$$
and the probability of striking from a zone that is not any of the four corners is:(3)$$p_{n4c}=\gamma 0+\left(1-\gamma \right)\frac{1}{12})=\frac{1-\gamma }{12}$$

We can obtain the simulated standard entropy H(γ) as a function of γ while taking into account that four zones (1, 4, 9, and 12) have probability $p_{4c}$ and the eight remaining zones have a probability $p_{n4c}$. Thus:(4)$$H_{sim}\left(\gamma \right)=-\frac{4p_{4c}\ln\left(p_{4c}\right)+8p_{n4c}\ln\left(p_{n4c}\right)}{\ln(12)}$$

The solid line in Figure 6 corresponds to the analytical expression of $H_{sim}\left(\gamma \right)$. We can observe the accordance with the numerical simulations obtained with the Monte Carlo simulation. Note that the analytical expression only gives information about the average values of the standard entropy but not on the sizes of the corresponding error bars.

### 3.4. Spatial Entropy

Now, we can quantify the entropy of the striking positions using a complementary indicator: the spatial entropy $H_{s}$. As explained in the Materials and Methods Section, spatial entropy reveals additional information when compared with its most extended counterpart, the standard entropy *H*. The main advantage of $H_{s}$ is that we can infer whether the spatial distribution of strokes is related to clustered, ordered, or random patterns. Figure 7a shows the spatial entropy of matches when the strokes have been separated according to won and lost points. As in the case of the standard entropy, the spatial entropies associated with the striking positions of the won and lost points overlap. Interestingly, we can observe how, in both cases, the values of the spatial entropy are bounded in the interval $0.3<H_{s}<0.8$, indicating that the spatial distribution of the striking positions lies between a random distribution ($H_{s}$ close to one) and a clustered distribution ($H_{s}$ close to zero). Furthermore, by no means the striking positions were distributed regularly.

In Figure 7b,c, we analyzed the influences of the score and the standing in the match on the spatial entropy. Specifically, Figure 7b shows the entropies of the positions when a player was leading the score (green) and when she was losing (red). We do not see essential differences between both distributions. In Figure 7c, the distribution in green refers to the initial points (when none of the players had reached 11 points), and the distribution in red is for final points (at least one of the players was above 11 points). Again, we do not see significant differences between the distributions.

Finally, we checked whether the mean values of the distributions shown in Figure 7 are equal using the paired Student’s *t*-test. Table 1 summarizes the averages of the distributions and their corresponding *p*-values. We can observe how in all three cases, the *p*-values are less than 0.5; however, the differences between the average values are quite low, suggesting we should be cautious about the interpretation of the results.

## 4. Discussion

The use of entropy allows bridging the gap between technical and tactical performance indicators in badminton. Therefore, performance can be related in an integrated approach to how and where the actions occur. This information is extremely important to winning points in a sport requiring high levels of anticipation, speed, reaction time, and responsiveness adapted to the opponent’s actions [16,17].

This study was focused on the entropy of the hitting positions of badminton players using two different metrics: the standard entropy (*H*) and the spatial entropy ($H_{s}$). Both metrics are related to the dispersion and randomness of the positions from which a player hits the shuttlecock. The analysis of the standard entropy revealed that the striking positions follow quite random distributions, a fact that was confirmed by the values of the spatial entropy. However, the interpretation of the spatial entropy suggests that the striking distribution lays between a situation in which all positions are clustered (leading to values of $H_{s}$ close to zero) and a purely random distribution (with $H_{s}∼$ 1). In that sense, the fact that all spatial entropies are bounded by the interval [0,1] supports the idea that the striking positions do not follow regular patterns, which would lead to values of spatial entropy close to 2. We have designed a model that aims to capture this kind of pattern. We simulated matches while allowing the players to choose the striking position between hitting from the four corners or hitting from random positions. The preference for either strategy was tuned by a control parameter γ, and the results showed that the simulated model with intermediate values of γ had a standard entropy similar to that reported in real matches, indicating a balanced combination of both striking patterns.

We also calculated the standard entropies of the striking positions for the won and lost rallies. We obtained similar results for both conditions; thus, we cannot affirm that entropy (in the way it is computed) is a successful performance indicator. Spatial entropy led to similar results even when two other filters were considered: (i) distinguishing between leading the score or being behind and (ii) separating points at the beginning/end of the match. None of these filters showed significant differences.

This finding reinforces the idea of stable spatial strategies during matches with both players using the zones in a similar way [9]. However, further research should investigate the inter-related effects of playing patterns and space used, with a clear approach that focuses on performance variability during each rally based on the quality of opposition and players’ characteristics (i.e., handedness).

On the other hand, it is worth mentioning the results concerning the interplay between the specific striking position and the region where the shuttlecock was sent. Specifically, when a player hit the shuttlecock from a region *i*, we calculated the standard entropy of the position *j* where the shuttlecock was sent. In this way, we obtained an entropy associated with each of the twelve regions of the pitch. We observed that regions 10 and 11, placed at the central back part of the pitch, were those with the lowest entropy; regions 9 and 12, also at the back, but both sides of the pitch, were those with the highest entropy. A closer inspection of this fact led to the following conclusion: hitting from regions at the sides of the back of the pitch leads to high entropy in the majority of matches, whereas hitting from the central back positions results in a diverse distribution of spatial entropies, with a low average value. As recently identified, the presence of an opponent modifies actions and subsequent reactions [18], so the use of zones 9 and 12 may open up more space in the opponent’s court, allowing more significant variability of hitting responses. However, when the players are more affected by the presence of the opponent in central zones, the hitting response is more predictable and then has less variability (entropy).

Finally, the current study has some limitations that must be addressed in future research. On the one hand, the striking positions should be studied in a multivariate approach while controlling for temporal (e.g., rally time, frequency between strokes, or rest time), technical (e.g., type of serve and striking technique), tactical (e.g., sequence of strikes), and player-related (e.g., handedness, ranking, or style of play) variables. On the other hand, the context of play needs to be accounted for by considering the score-line, the number of sets, the interval during the set, the stage of the competition, or the quality of opposition as crucial factors for the striking positions and the performance.

We believe that our results could be a starting point for using complex scientific metrics for understanding badminton, and more generally, racket sports. Furthermore, if statistics were focused on specific players, they could be used to prepare for a match by analyzing the striking patterns of future opponents. The current findings are of great relevance to the coaching staff and the players. In particular, the identification of lower or greater spatial entropy in elite badminton may be used to better design training drills focused on generating more unpredictability during rallies and then increasing the probability of being better at adapting to changes of an opponent’s hitting position. In particular, zones 9 and 12 should be considered as relevant zones when designing specific tasks involving more space in the opponent’s half during patterns of 3, 4, 5, or 6 strokes.

The following abbreviations are used in this manuscript:

## Figures and Tables

**Figure 1 entropy-23-00799-f001:**
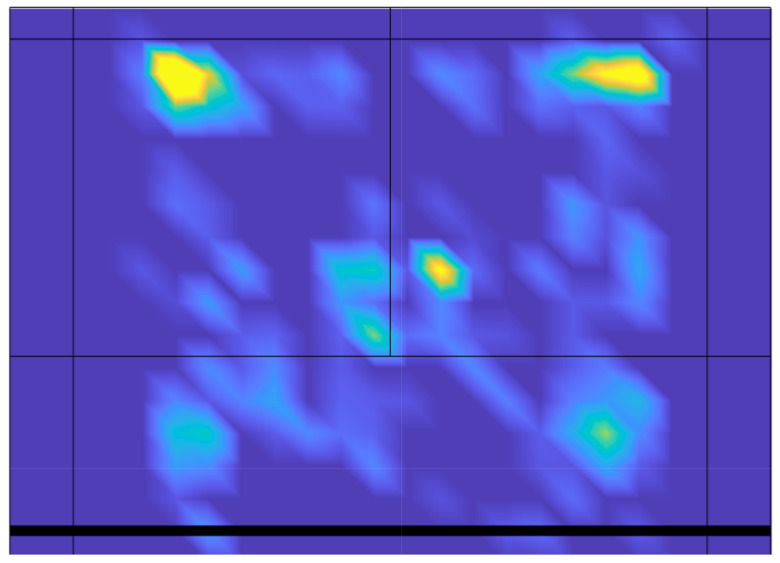
Heat map of the *X* and *Y* striking positions on the badminton court. We plot the average of 259 matches played by 63 female players.

**Figure 2 entropy-23-00799-f002:**
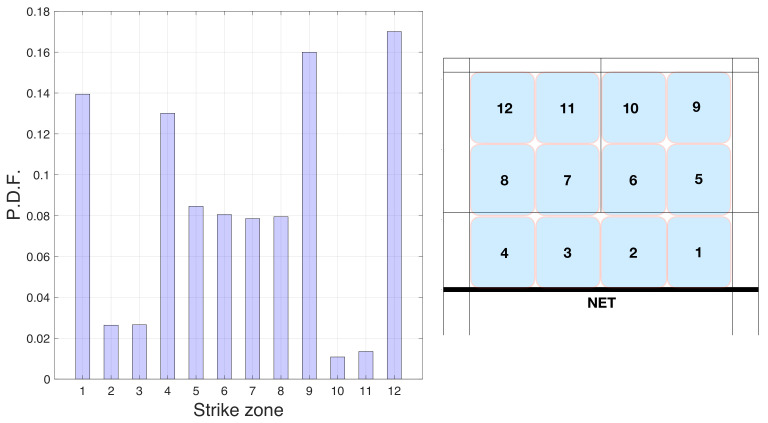
In the left plot, we have the probability distribution function (PDF) showing the percentages of strokes performed in the 12 pre-defined zones of the court (see the right plot for the location of each zone).

**Figure 3 entropy-23-00799-f003:**
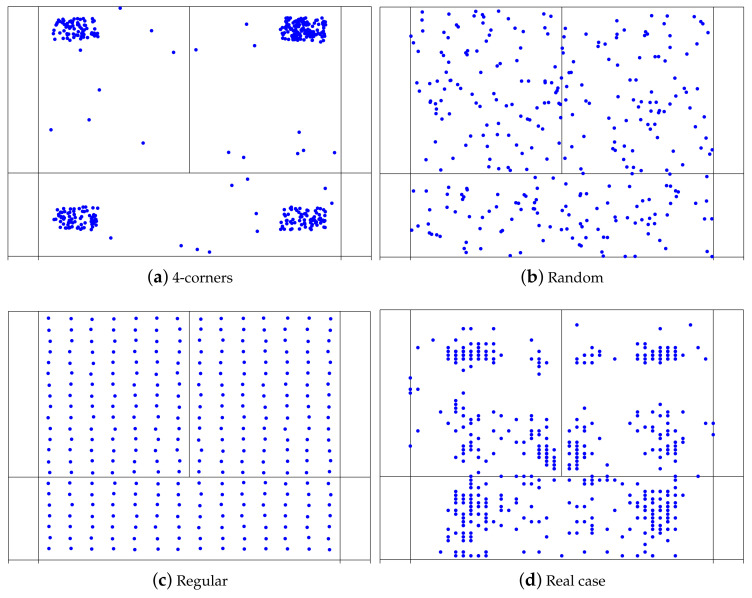
(**a**) Spatial entropy of the striking position. Plots (**a**–**c**) have been generated numerically, whereas (**d**) is a real case. Specifically, in (**a**) the stroke positions were biased toward the four corners of the court. The corresponding spatial entropy is 0.462. In (**b**), the strokes were randomly distributed, leading to $H_{s}=0.997$. (**c**) It corresponds to the case where the strokes are placed following a regular lattice, maximizing the distances between them and leading to $H_{s}=1.985$. Finally, (**d**) corresponds to the striking position of Carolina Marín playing against Chen Xiaoxin during the China Open 2018. In this case, $H_{s}=0.599$.

**Figure 4 entropy-23-00799-f004:**
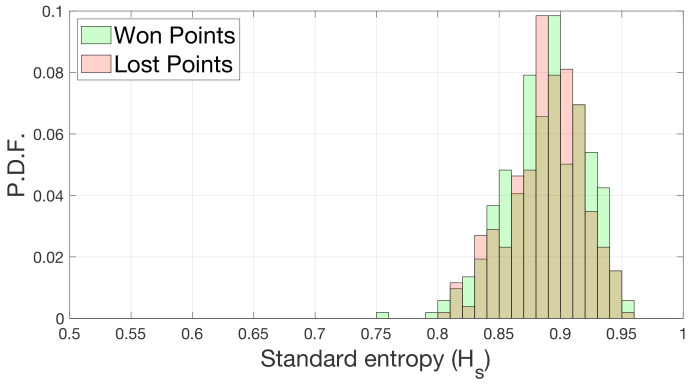
Probability distribution function (PDF) of the standard entropy (*H*) for won and lost points. Entropies have been calculated per match. In green are the *H* values of the positions from which the winning players hit. In red are the *H* values of the positions from which losing players hit. The shaded areas reflect the overlap of the two distributions.

**Figure 5 entropy-23-00799-f005:**
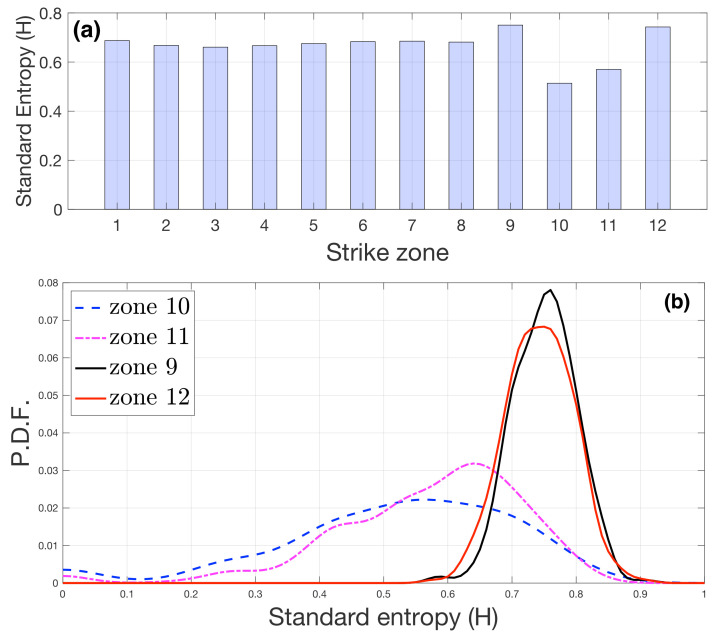
Probability distribution function (PDF) of the standard entropy (*H*) for the receiving zones. (**a**) Average of the receiving zones’ entropy over all matches. (**b**) PDF of the receiving entropy for the most different zones: 9, 10, 11, 12.

**Figure 6 entropy-23-00799-f006:**
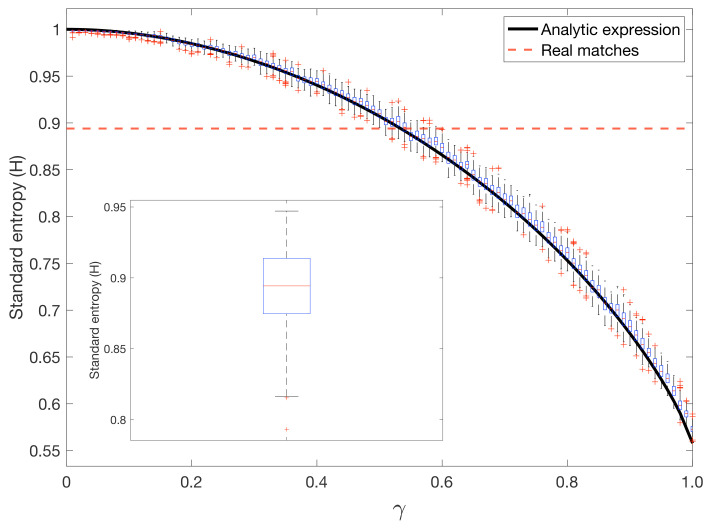
From random to four-corner striking patterns. A parameter γ controlled the amount of randomness of the striking positions. For γ=0, the hitting positions were purely random, whereas a player was always striking from the four corners when γ=1. For each value of γ, each of the N=259 matches was simulated, while maintaining the same number of strokes. Each match was simulated 100 times. On each box, the central mark indicates the median, and the bottom and top edges of the box indicate the 25th and 75th percentiles, respectively. The red dashed line corresponds to the average standard entropy of the real matches ${\bar{H}}_{real}=0.894$. The inset shows the boxplot corresponding to the distribution of the standard entropy of the real matches (median, 25th, and 75th percentiles and outliers). Finally, the solid black line corresponds to the analytical expression of the simulated Standard Entropy $H_{sim}\left(\gamma \right)$ (see Equation (Equation 4)).

**Figure 7 entropy-23-00799-f007:**
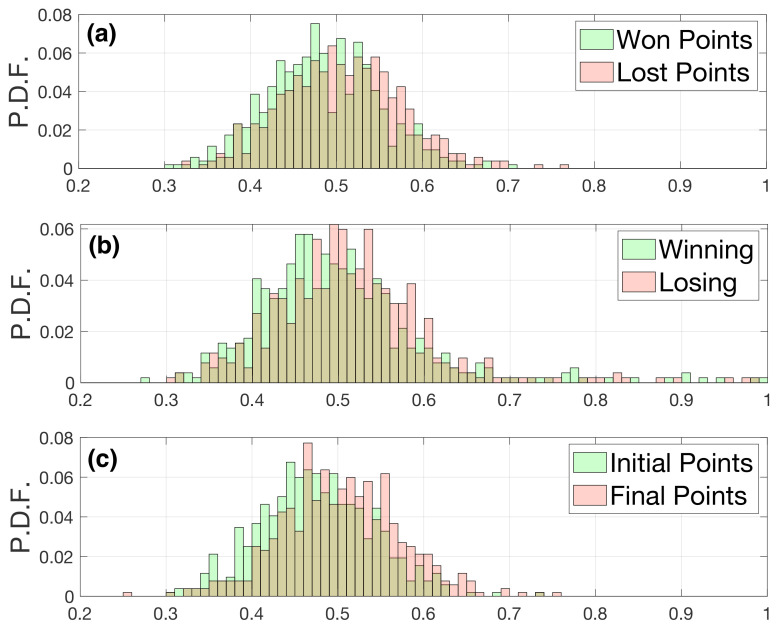
Spatial entropy $H_{s}$ for (**a**) won and lost points, (**b**) points from the leader or the player behind, and (**c**) initial points vs. final points. The green bars refer to the entropies of matches computed over the won (**a**), points while leading (**b**), or initial (**c**) points during a match, whereas red bars correspond to lost points (**b**), points while being behind in score (**b**), and final points (**c**). See main text for details. The shaded areas reflect the overlap of the two distributions.

**Table 1 entropy-23-00799-t001:** Results of the paired Student’s *t*-test of the spatial entropy $H_{s}$ distributions of Figure 7.

Situation	(1) Mean $H_{s}$	(2) Mean $H_{s}$	*p*-Value
Won (1) vs Lost (2) points [ Figure 7a]	0.484	0.507	6.051 × 10${}^{-18}$
Winning (1) vs Losing (2) [ Figure 7b]	0.486	0.520	0.006
Initial (1) vs Final (2) points [ Figure 7c]	0.474	0.503	2.90 × 10${}^{-35}$

## Data Availability

Data will not be publicly available, but any person interested in can require it writing the corresponding author via email.

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
