# Peer review of "Entropy of Badminton Strike Positions"

_entropy, 2021, doi:10.3390/e23070799_

Round 1
Reviewer 1 Report
Please find my comments attached.

Author Response
In my opinion the paper is placed in the context of the pertinent literature and is interesting for the audience of the journal, nevertheless I think that it needs clarification is several points related with the presentation of the data. This will make the paper more understandable even for the non specialist. I recommend reconsideration after major revisions along the following points:
- In equation (1), the inclusion of the limits $_{i=1}^{12}$ of the sum is important to clarify that this expression is used to compute the entropy of the probability density that includes the data organized by zones.
We absolutely agree with the referee, we have included the limits.
- In line 80, is it better said “from zone i” instead of “from position i”?
We have replaced position by zone in the manuscript.
- Line 135, what is the “probability distribution of the standard entropy” presented in figure 4? An equation defining this probability function should be included for the sake of clarity.
We are very sorry that we did not explain well the results of standard entropy. We have included some paragraphs to improve the clarity of the manuscript.
- The presentation of data in figure 4 is confusing because of various points:
Trying to answer this question, to verify that the entropy values were being calculated per match, with a total of 259 entropy values per zone, we realized that winning and losing entropy values were calculated including the position of the shuttlecock when hitting the net our going outside the court. We recalculated again all entropies without considering these positions, which leaded to a set of new figures.
we appreciate the reviewer's comment and we hope that with the new paragraph the idea of these figures have been clarified.
- a) I assume that the probabilities used in the calculation of H come from the number of rallies. If so, the authors should include this in the list of data. How many rallies are they considering?
Entropies are calculated per match. We have mentioned this fact explicitly in the revision. On that sense, the number of rallies is not relevant.
- b) In connection with my previous comment, the formula to calculate H is not presented and it is not the same as Eq. (1). The authors should present the equation used for these calculations.
The equation to obtain the results of Fig. 4 is Eq(1), where strokes performed when winning a point have been separated from losing a point. We have mentioned this fact in the text.
- c) Are the number of events considered in the red bars the same as in the green ones? (because a lost point for one player is won by the other). Or are the authors considering one side of the game? This comment is in connection with the previous one in a).
The number of winning and losing strokes is not exactly the same, but it is similar.
- d) What are the brown bars?
Brown bars are just indicating the overlap of the two PDFs. We have clarified this point in the text.
- e) In line 140 the authors conclude that higher entropies are indicative of more random strokes, but it seems that these higher values are indicative of more ordered ones. To clarify this point they should perform an analysis similar to that in figure 2, but for the standard entropy, to argue how they distinguish between both scenarios.
Higher values of entropy are closer to 1, which is the highest possible value. The more entropy we have in a system, the closer we are to a random distribution.
- f) Can the bars be interpreted as follows? The second green bar from right to left indicates that 2% of the rallies (considering the side of winner of the point) have entropies in the range 0.94-0.95? The authors should include
The interpretation is correct. However, not rallies but matches. We have clarified this point in the revised version.
- Data presented in figure 6 are also confusing. My comments are similar to that presented in items a)-f) of the previous point (for figure 4).
We believe that with the previous explanations and changes, the problem is solved.
- Similar comments to a), b), c), d) and f) from Figure 4 apply to Figures 7 and 8.
We believe that with the previous explanations and changes, the problem is solved.
- In line 190 and 193 the authors claim that they do not see significant differences between the distributions in Figures 7 and 8, respectively, but I could not understand why. The authors should clarify this point.
Despite there are slight differences in the average values of the distributions, they are statistically significant, as indicated by a t-test.
Reviewer 2 Report
The article is an interesting analysis, but many points about the proposed model are obscure and also some other things are not clear. The quality of the presentation must be improved. I ask for major revisions. My main comments are the following.
MAIN COMMENTS
- Fig. 4 is the p.d.f. of the standard entropy of the stroke hitting positions calculated on RALLIES or MATCHES? Put another way, if the histogram were not normalised to get the p.d.f., would the sum of all bars be 259, the number of matches? This is not clear from the text. Please, clarify.
- Section 3.3.
- Since all the considerations are made for the position FROM WHERE strokes are hit and this depends on what the opponent plays, using the terminology "strategy" is confusing: "strategy" makes the reader think of a conscious decision by the hitting player, while this does not seem the case. Please clarify.
- The simulation of the model (l. 157--163) is not clear. Explain better how it is performed.
- L. 167 "hitting TO the four-corners" or "hitting FROM the four-corners"? TO is very confusing.
- L. 169--171 Finding an entropy value for the real case (H=0.89) which lies between 1 and 0.556 is almost automatic. Deriving from this value a corresponding value of gamma just by inverting the curve is not that significant in my opinion.
- Fig. 5. Drawing the real matches value with just the standard deviation as error bars is not that informative. Much better would be either plotting a boxplot with interquartile range and max and min values or error bars representing 95% of the real distribution.
- Given the model, the probabilities pi of the hitting positions can be calculated explicitly. Why is a Monte Carlo simulation actually needed?
- Fig. 5. Explain how you simulate one match.
- L. 189--190 and Fig. 7. Apart from visual assessment, a quantitative measurement of the difference between the two distributions (one possibility is the Kullback-Leibler divergence but others are possible) and/or a test for equality between discrete distributions would give a more objective evaluation. Please provide at least one such measurement/test.
- L. 193--194 and Fig. 8. Same as in previous comment.
- L. 218--221 This conclusion is weak. As said before, saying that players will find themselves between the two extremes of hitting from the four corners and hitting from random places is somewhat obvious and a model would not even be needed. If the authors want to stress that the entropy value of real matches lies strictly (i.e., considering error bars) between the two extremes, this should be clarified.
- Some further information on the data set (section 2.1) would be interesting, if data are available:
- are both right-handed and left-handed players represented?
- what system has been used to record the hitting points?
- where is the data set from?
- Check English and typos.
Author Response
Fig. 4 is the p.d.f. of the standard entropy of the stroke hitting positions calculated on RALLIES or MATCHES? Put another way, if the histogram were not normalised to get the p.d.f., would the sum of all bars be 259, the number of matches? This is not clear from the text. Please, clarify.
Trying to answer this question, to verify that the entropy values were being calculated per match, with a total of 259 entropy values per zone, we realized that winning and losing entropy values were calculated including the position of the shuttlecock when hitting the net our going outside the court. We recalculated again all entropies without considering these positions, which leaded to a set of new figures.
Concerning the Reviewer’s question, the entropy is calculated per match, therefore, we have 259 values for each distribution. We have explicitly mentioned this fact in the Figure’s caption.
- Section 3.3.
- Since all the considerations are made for the position FROM WHERE strokes are hit and this depends on what the opponent plays, using the terminology "strategy" is confusing: "strategy" makes the reader think of a conscious decision by the hitting player, while this does not seem the case. Please clarify.
We have removed the term strategy both in Section 3.3 and the Discussion. We now refer just to the “striking patterns”, without any reference about the intention or strategy of the player.
- The simulation of the model (l. 157--163) is not clear. Explain better how it is performed.
We have extended the explanation of the model in the manuscript.
- L. 167 "hitting TO the four-corners" or "hitting FROM the four-corners"? TO is veryconfusing.
The Referee is right. We have corrected the expression (here and in the rest of the text).
- L. 169--171 Finding an entropy value for the real case (H=0.89) which lies between 1 and 0.556 is almost automatic. Deriving from this value a corresponding value of gamma just by inverting the curve is not that significant in my opinion.
The main purpose of plotting the value of real matches was to understand whether the entropy values of real matches was close to any of the two situations considered in the model (i.e., striking from the 4 corners or from random positions). Our results show that it is not the case. As we mention in the text, a model with gamma=0.55 would lead to similar values of the entropy. However, there are a diversity of variables that affect the striking positions and are not considered in our toy model, therefore we cannot say that striking positions are given by (just) a combination of both strategies. We have clarified this point in the text.
- Fig. 5. Drawing the real matches value with just the standard deviation as error bars is not that informative. Much better would be either plotting a boxplot with interquartile range and max and min values or error bars representing 95% of the real distribution.
Following the Referee’s suggestion, we have re-drawn Fig. 5 using a bloxplot representation instead of the average and standard deviation. On each box, the central mark indicates the median, and the bottom and top edges of the box indicate the 25th and 75th percentiles, respectively. We have plotted the values corresponding to the real matches in an inset (in order to avoid overlapping with the simulated ones).
- Given the model, the probabilities pi of the hitting positions can be calculated explicitly. Why is a Monte Carlo simulation actually needed?
The purpose of the Monte Carlo simulation is to capture the stochastic nature of the game. Using this kind of simulation, we adjusted the number of strokes to the one of the real matches, which allowed us to calculate the error bars, also giving useful information about the model. Note that, analytically, we can just extract the average value of the entropy. After reading the Referee’s suggestion, we decided to include the value of the entropies obtained analytically.
- Fig. 5. Explain how you simulate one match.
We have included a paragraph describing, step by step, the simulation of all matches.
- L. 189--190 and Fig. 7. Apart from visual assessment, a quantitative measurement of the difference between the two distributions (one possibility is the Kullback-Leibler divergence but others are possible) and/or a test for equality between discrete distributions would give a more objective evaluation. Please provide at least one such measurement/test.
- L. 193--194 and Fig. 8. Same as in previous comment.
We appreciate the reviewer's comment, with the new results we think that a good way to make a measure of the mean values of the distributions is using a paired t-test. We have introduced the results in a paragraph in the manuscript.
- L. 218--221 This conclusion is weak. As said before, saying that players will find themselves between the two extremes of hitting from the four corners and hitting from random places is somewhat obvious and a model would not even be needed. If the authors want to stress that the entropy value of real matches lies strictly (i.e., considering error bars) between the two extremes, this should be clarified.
Concerning this point, it was not so obvious (at least for us) that the results of the model would show that the level of entropy associated to the hitting points is compatible to a situation balancing the four-corner and the random extreme situations. In fact, we were expecting a level of entropy closer to the four-corner case, as suggested by Fig.3 (left plot). Note that we are not saying that a badminton player is just conditioned by these two particular patterns (four-corners vs random), since there is a diversity of other factors influencing the striking position (the moment of the game, the score, fatigue, etc…). On that sense, we consider our toy model as a starting point where other variables could be added in more advanced versions. We have clarified this point in the text.
- Some further information on the data set (section 2.1) would be interesting, if data are available:
- are both right-handed and left-handed players represented?
Yes, the sample included both handedness for women players.
- what system has been used to record the hitting points?
We have included this paragraph in the manuscript at the end of the section 2.1.
“Data was gathered using the Dartfish video analysis software (Dartfish, Friburgo, Switzerland) by two performance analysts (with five years of experience as elite badminton coaches) trained to this observational task. Data reliability was tested for inter- and intra-observer match analyses with very good values (Kappa, ICC, Pearson’s correlation and TEM) using 2 random matches.”
- where is the data set from?
We have included this paragraph in the manuscript at the end of the section 2.1.
“The sample was composed by 259 women’s matches (video recorded from the World Badminton Federation) played during the 2015-2019 World Badminton Super Series, the 2016, 2018 European Championship, the 2016 Olympic Games, and the 2015, 2017, 2018 and 2019 World Championship.”
- Check English and typos.
We have checked the English again and corrected all typos. Thank you for pointing to this issue.
Reviewer 3 Report
Dear authors
I have carefully reviewed your manuscript. I consider that it is interesting for the readers and it is very well written. However, I have a number of suggestions before I support its publication.
Please modify the keywords. They should not be repeated with those of the title. Also, I recommend including 2-3 more.
On the other hand, the abstract may begin with the aim of the study. The problem is that in the introduction does not have a well-written aim either. Please correct this aspect.
Do authors think that Figure 1 is well positioned in this part of the manuscript? I consider that it may be moved of position (or eliminated directly). It must also be correctly cited during the text.
A section of limitations may be included. Furthermore, the authors may develop the practical applications for badminton coaches.
Author Response
- Please modify the keywords. They should not be repeated with those of the title. Also, I recommend including 2-3 more.
We have modified the keywords, using: Shannon entropy; spatial entropy; racket sports; match analysis; performance.
- On the other hand, the abstract may begin with the aim of the study. The problem is that in the introduction does not have a well-written aim either. Please correct this aspect.
We have added the aim of the study in the first paragraph of the abstract and at the end of the introduction.
- Do authors think that Figure 1 is well positioned in this part of the manuscript? I consider that it may be moved of position (or eliminated directly). It must also be correctly cited during the text.
We agree with the reviewer that it is an odd position for Figure 1, but we think the figure is a starting point for understanding the four-corner pattern and subsequent entropy results. If that were possible, we would like to keep the figure in this position.
- A section of limitations may be included.
We agree have included a paragraph in the Discussion about the limitations of our study.
- Furthermore, the authors may develop the practical applications for badminton coaches.
We have included a paragraph in the Discussion section suggesting some practical applications. Despite it is not mentioned in the manuscript, we are currently working on the development of an application for badminton coaches.
Round 2
Reviewer 1 Report
I think that the authors have improved significantly the manuscript and have properly taken into account my comments. Thus I recommend publication after a minor spelling English revision.
Author Response
Thanks for your comments, we have done a detailed review of the English language.
Reviewer 2 Report
I believe that the last modifications to the article have increased its quality significantly.
There are still many English typos (l. 159 are equals, page 9 as as function, l. 281 where the all positions, l. 305 where, to point out just a few).
Correct also the following:
L. 212: 5. If ran1(i) > γ (i.e., four-corner random pattern)
Author Response

(The authors gave the same response as above.)

Reviewer 3 Report
Dear authors
I think you have done a great job improving the quality of this paper. the paper is ready to support its publication
Author Response

(The authors gave the same response as above.)
